# MRI Appearance of Focal Lesions in Liver Iron Overload

**DOI:** 10.3390/diagnostics12040891

**Published:** 2022-04-02

**Authors:** Anna Pecorelli, Paola Franceschi, Lorenzo Braccischi, Federica Izzo, Matteo Renzulli, Rita Golfieri

**Affiliations:** Department of Radiology, IRCSS Azienda Ospedaliero Universitaria di Bologna, 40138 Bologna, Italy; paola.franceschi@studio.unibo.it (P.F.); lorenzo.braccischi@studio.unibo.it (L.B.); fedeizzo.92@gmail.com (F.I.); matteo.renzulli@aosp.bo.it (M.R.); rita.golfieri@unibo.it (R.G.)

**Keywords:** adenoma, extramedullary hematopoiesis, hepatocellular carcinoma, liver iron overload, magnetic resonance imaging

## Abstract

Liver iron overload is defined as an accumulation of the chemical element Fe in the hepatic parenchyma that exceeds the normal storage. When iron accumulates, it can be toxic for the liver by producing inflammation and cell damage. This can potentially lead to cirrhosis and hepatocellular carcinoma, as well as to other liver lesions depending on the underlying condition associated to liver iron overload. The correct assessment of liver iron storage is pivotal to drive the best treatment and prevent complication. Nowadays, magnetic resonance imaging (MRI) is the best non-invasive modality to detect and quantify liver iron overload. However, due to its superparamagnetic properties, iron provides a natural source of contrast enhancement that can make challenging the differential diagnosis between different focal liver lesions (FLLs). To date, a fully comprehensive description of MRI features of liver lesions commonly found in iron-overloaded liver is lacking in the literature. Through an extensive review of the published literature, we aim to summarize the MRI signal intensity and enhancement pattern of the most common FLLs that can occur in liver iron overload.

## 1. Introduction

Iron overload is defined as the accumulation of iron in the body, which can be local or systemic, and is classified as primary when it is caused by genetic errors or secondary, when it depends on acquired pathogenetic conditions [1]. Although the liver is the main iron storage of the human body, when hepatic iron deposition exceeds the normal level, it can cause liver damage, leading to cirrhosis and eventually to hepatocarcinoma (HCC) [2]. The diagnosis of liver iron overload and its quantification are pivotal to drive the best treatment approach and to assess the treatment response, respectively. To date magnetic resonance imaging (MRI) is considered the best non-invasive technique for detection and quantification of iron levels in patients with iron overload disease [3,4,5].

Iron accumulation in the liver can cause a loss of parenchymal signal intensity. In MRI sequences, intensity is defined as the shade of grey of a particular tissue (or lesion) in comparison to other tissues. In case of FLLs, intensity (hyper, hypo or iso) is defined in comparison to that of the background liver parenchyma. Thus, for example, a malignant lesion in a normal liver will show slight hyperintensity on T2 sequences, while the same lesion will appear more hyperintense in an iron-overloaded liver, raising the suspicion of benign lesions. This can potentially create a problem of differential diagnosis of FLLs as they can appear with a relatively different signal intensity on MRI sequences. The present review critically evaluates the MRI findings of some of the most common hepatic focal lesions in patients with liver iron overload.

## 2. Iron in the Human Body: Metabolism, Overload and Pathogenetic Mechanisms

### 2.1. Iron Metabolism

Iron is a vital element for many cellular activities due to its capacity to participate in oxidation-reduction reactions, but when in excess it can also generate damage with oxygen radicals, thanks to its ability of donating and accepting electrons [1,6]. Each day, an adult assumes and loses 1–2 mg of iron, which is absorbed by the duodenum and then deposited into tissue storage (3–4 g), such as the plasma transferrin, tissue and myoglobin pools [7]. The hormone called Hepcidin regulates human iron storage levels bounding the iron exporting pump ferroportin and determining ferroportin internalization and degradation [8,9]. Inactivation of this serum protein leads to hypoferremia and cellular iron retention, especially in macrophages and hepatocytes [1,6,7,8]. In cells, transferrin binds iron because free iron is highly reactive and generates reactive oxygen species (ROS) [1]. In the condition of hyperferremia, the transferrin saturation is higher, hence non-bounded iron promotes cell death and tissue damage [6].

### 2.2. Iron Overload: Primary and Secondary

Considering genetic disorders, the most common cause of iron overload is hereditary hemochromatosis (HH), which is divided in two groups. The first group is linked to the polymorphism C282Y, the so called “high iron Fe” (*HFE*) gene, whereas the second group includes all the non-HFE genetic syndromes [1,5,10]. Non-HFE hereditary hemochromatosis is mostly associated with pathological mutations of proteins, which orchestrate iron homeostasis, such as hepcidin (*HAMP*), ferroportin (*FPN*), hemojuvelin (*HJV*) and transferrin receptor 2 (*TFR2*) [1]. HFE mutations can be found in over 80% of patients with hemochromatosis [9,11]. HH remains clinically silent until late adolescence in non-HFE disorders and until middle age in HFE-associated hemochromatosis [12,13]. Red blood cell disorders are involved in most of the cases of secondary hemosiderosis. In fact, long-term blood transfusions, necessary in case of hereditary anemias or bone marrow diseases, such as myelodysplasia, result in increased iron reserves [1]. Moreover, some syndromes, such as hemoglobinopathies, interfere in hepcidin expression, causing increased iron intestinal uptake. Other causes of secondary hemochromatosis comprise, all the commonest chronic liver diseases, such as viral hepatitis and alcoholic and non-alcoholic fatty liver disease (NAFLD). In these latter cases the mechanisms that contribute to liver iron overload are multiple. Other causes of secondary hemosiderosis, such as porphyria cutanea tarda, alloimmune neonatal hemochromatosis or African siderosis are rarer [1]. Symptoms vary depending on which organs are most affected by iron overload; in non-HFE hemochromatosis, heart and endocrine glands are frequently affected, in patients with HFE mutation, the liver is mostly damaged, whereas in secondary hemosiderosis, iron overload hits the reticuloendothelial system (bone marrow and spleen) and, in part, the liver. ROS-induced damage determines different syndromes in each tissue, such as diabetes and hypothyroidism in endocrine glands and heart failure, joint injury (arthritis) and fibrosis in the liver [1,14].

### 2.3. Liver Iron Overload and Pathogenetic Mechanisms

All the hereditary and some acquired iron disorders lead to a deficiency of hepcidin, compromising regulation of iron uptake and distribution [6,15]. Hepcidin is the key regulator of iron metabolism and is produced and secreted predominantly by hepatocytes and macrophages on their membranes [2,7,8]. Several hepcidin regulators control its synthesis, and most of those are included in the mutated genes, which lead to hemochromatosis (e.g., *HFE, TfR2, HEF2, HAMP* etc.). Multiple signals orchestrate iron homeostasis, such as chronic inflammation, iron plasma levels, erythroid activity, oxygen tissue need, diet, intestinal disorder, etc. [7,8]. Liver is the commonest tissue of iron deposition and toxicity. In fact, the iron-mediated tissue damage generates fibrogenesis, by activating hepatic stellate cells and portal myofibroblasts causing cirrhosis [1,2,6]. Moreover, production of ROS by iron toxicity induces specific DNA damages which can directly influence hepatocarinogenesis [1]. For this reason, early diagnosis and quantification of iron levels by a non-invasive method, such as MRI, allows the fastest possible treatment of patients with iron overload and is crucial for their prognosis [3,5].

## 3. Imaging Diagnosis of Liver Iron Overload: Detection and Quantification Using MRI

Different non-invasive methods have been proposed to evaluate the liver iron concentration during the last decades. MRI is recognized as the reference non-invasive method for the assessment and quantification of liver iron overload. It also allows the monitoring of the response to treatment of iron overload disease and prevent complications [16,17]. The liver iron deposits introduce inhomogeneities in the main magnetic field due to the superparamagnetic effect of the iron ions. This leads to the reduction of the relaxation time of hydrogen nuclei and signal intensity with increasing amounts of iron [17,18].

### 3.1. Qualitative Detection Using MRI

It is possible to detect hepatic iron accumulation by comparing in-phase and opposed-phase gradient echo protocols. In-phase protocols are more sensitive to the T2* decay produced by iron deposits, which causes the loss of signal intensity in the affected tissues on in-phase gradient echo protocols compared to opposed-phase ones. A limitation of this method is the coexistence of liver steatosis and iron overload, due to signal intensity decay on the opposed-phase protocols that can cover T2* decay on the in-phase protocols [17,19]. Another qualitative method to evaluate the iron accumulation is comparing the reduced signal intensity in affected tissue to other unaffected tissue using T2-weighted fast spin-echo sequences. However, qualitative methods are not able to evaluate the degree of iron overload so quantitative methods are required [3].

### 3.2. Quantification of Liver Iron Overload Using MRI

Several studies in the literature have shown the correlation between liver iron concentration (LIC) values and MRI measurements [20,21]. In the last years, two advanced methods have been used: relaxometry and signal-intensity ratio [22].

#### 3.2.1. R2 and R2* Relaxometry

The relaxometry method is based on calculation of the T2 time constant, on spin-echo sequences, and T2* time constant on the gradient-echo sequences; both are estimated from MRI signal loss in images acquired at several TEs. In order to increase the range of values, we use T2 and T2* inverses, respectively R2 = 1000/T2 and R2* = 1000/T2*, which increase in the presence of iron [16,22]. The exact mechanism of the R2 increase is not clear, two processes have been proposed: on one hand, the microscopic inhomogeneities introduced in the magnetic field by iron; on the other hand, the proton exchange that occurs between tissue water and protons bounded to iron-containing proteins. A monotonic nonlinear correlation has been shown between LIC and the R2 increase. The R2* increase in gradient-echo images is due to the microscopic magnetic field inhomogeneities, which result in rapid spin offset. The correlation between R2* increase and LIC can be fitted linearly [3]. The R2 relaxometry method was developed by St. Pierre et al., marketed as FerriScan and approved by the US Food and Drug Administration (FDA) for 1.5 T scanners. The acquisition protocol consists of five T2 weighted multislice single spin-echo (SSE) free-breathing pulse sequences with constant TR of 2500 ms and increasing TE (from 3 ms to 6 ms to 18 ms), slice thickness of 5 mm, flip angle of 90°, matrix size of 256 and FOV between 350 and 400 mm. St. Pierre et al. selected an ROI bounded at the right hepatic lobe on the largest axial slice of the liver and modeled a bi-exponential equation to the image signal intensities decay measured at each echo-time to estimate R2 value. Furthermore, they developed a calibration curve relating the R2 values to the LIC [23]. The main limitation of this method is the non-linear relationship between the R2 measurements and the hepatic iron concentration, which results in sensitivity loss to iron concentrations over 20 mg/g of dry weight. In addition, the technique is not able to evaluate the simultaneous presence of liver steatosis and iron overload as fat signal is refocused when using spin echo sequence, unless fat saturation is applied and it requires a long acquisition time (about 20 min) that leads to increasing breath artifacts and delayed results because of centralized data analysis [5]. To overcome this limitation, a more recent variant of the St. Pierre method with a reduced TR of 1000 ms has been proposed. This method reduced the acquisition time not affecting the accuracy and precision of LIC measurements [24]. R2* relaxometry proposed by Wood et al. is based on GRE sequences with multiple breath holds, with TR of 25 ms, TE every 0.25 ms from 0.8 to 4.8 ms, slice thickness of 15 mm, flip angle 20°, matrix size of 64 × 64 and FOV of 48 × 24 mm. They calculated R2* values on a ROI drawn on a single mid liver section excluding hilar vessels to obtain an R2* map. Therefore, Wood et al. found the correlation between liver iron content and R2* with the equation [Fe] = 0.202 + 0.0254 R2* [25]. However, more recently, other R2* relaxometry techniques have been proposed, based on single breath-hold multiecho GRE sequences with different calibration curves for the calculation of R2* values and hepatic iron concentration, but there has not been univocal consensus on the recommended protocols [26,27].

R2* relaxometry overcomes some of the limitations of R2 method because of its shorter acquisition time, the reduction of breath artifacts due to the single breath-hold sequences and its ability to simultaneously quantify liver steatosis and iron overload. Furthermore, it can be used both on 1.5 T and 3.0 T scanners that allow to detect a wider range of liver iron content. However, this technique has some limitations too. In fact, it is sensitive to external magnetic inhomogeneities, such as metal clips and air, and it is affected by great results’ variability depending on the adopted protocol, resulting in reduced reproducibility [3,5].

#### 3.2.2. Signal-Intensity-Ratio

The SIR method is based on either spin echo (SE) or gradient echo (GRE) sequences, and MR measurements are obtained by calculating the SI ratio between liver tissue and other reference unaffected tissues, usually the paravertebral muscle [16]. The most used SIR method was proposed by Gandon et al. It consists of five GRE sequences acquired during separate breath holds, with a fixed TR of 120 ms, slice thickness of 10 mm, FOV of 40 cm, matrix size of 256 × 128, flip angle of 20°and multiple increasing TEs to obtain different weighted images: T1, proton-density and T2*-weighted sequences [28]. More recently, the technique has been reproduced on 3 T scanners [29]. Acquisition protocols are available online for 1.0 T, 1.5 T, and 3.0 T scanners [30]. Iron overload reduces the signal intensity ratio with a non-completely linear correlation depending on different weighting [31]. The SIR-LIC curve can be approximately linearized depending on the protocol used. Moreover, a linear dependence of the logarithm of SIR on LIC has been demonstrated [29]. The signal intensity is measured with three ROIs drawn in the liver parenchyma, excluding vessels and one drawn in each paraspinal muscle, all about 1 cm^2^ in area. These measurements are repeated with the five MR sequences mentioned above and used to calculate five separate liver to muscle (L/M) ratios. The results are analyzed with the algorithm developed by Gandon et al. that is available on the website of the University of Rennes to estimate LIC [32]. SIR is the easier liver iron overload quantification method and has the simplest and open access post-processing, meaning that it is accessible and feasible on every machine in the world [22]. Nevertheless, the technique shows reduced sensitivity for severe iron accumulation (over 350 μmol/g of dry weight) and tends to overestimate mild and moderate iron deposition [5]. For high LIC concentration, an extension of the SIR method has been published [33]. Moreover, the Spanish Society of Abdominal Imaging (SEDIA) has implemented the SIR method by using only two echoes (4 ms and 14 ms) and a different mathematical formula to calculate LIC. SEDIA’s model provided a better correlation with R2* and with LIC measured on liver biopsy [20].

## 4. Focal Liver Lesion in Hepatic Iron Overload

In iron body excess with reticuloendothelial, parenchymal or mixed deposition pattern, the liver iron overload provides a natural source of contrast enhancement similar to that yield by intravenous administration of superparamagnetic iron oxide particles (SPIO). This makes FLL devoid of iron hyperintense relative to the surrounding hepatic parenchyma on MR sequences mostly affected by susceptibility artifacts. A problem of differential diagnosis between benign and malignant lesions then arises [17]. The most common FLL possibly encountered in the setting of liver iron overload, and their appearance at MRI are described.

### 4.1. Hepatocellular Carcinoma

It is well known that liver iron overload with or without concomitant cirrhosis is associated with an increased risk of HCC, in particular in case of parenchymal iron deposition pattern. This linkage was first highlighted in patients with HH in which HCC usually arises on underlying and pre-existing cirrhosis [34]. Subsequently, it was demonstrated that an increased risk of HCC is also present in secondary hemochromatosis induced by dietary iron overload, the metabolic syndrome and chronic anemias with inefficient erythropoiesis. In these conditions and in rare cases of HH, HCC develops without concomitant cirrhosis. This has led to the understanding that iron represents a risk factor for HCC both indirectly, because it induces chronic liver inflammation leading to cirrhosis then to HCC, and directly, through the formation of free oxygen radicals and consequent oxidative tissue damage, and through the suppression of the immune system [35]. Regardless of the underlying disease (primary or secondary hemochromatosis) and of the presence or absence of cirrhosis, the high iron-overloaded liver, hypointense on T1 in-phase weighted images (WI) and on T2 WI, provides a natural source of contrast for the detection of iron-devoid hence hyperintense HCC. Only 5 cases of HCC in patients with secondary hemochromatosis have been reported in the literature, and only 3 of them describe MRI features of the lesions (Table 1). Two out of these three patients presented hemochromatosis secondary to myelodysplastic syndrome [36,37], while the third had no underlying disease but occupational exposure (coalminer) [38]. HCC always appears hyperintense on T1 and T2 WI relative to the liver parenchyma. DWI sequences were acquired only in one case and showed hyperintensity of the lesion [37]. HCC in primary hemochromatosis with liver iron overload, with or without concomitant cirrhosis, has been more widely described. In case of high liver iron concentration, it is usually defined as a focal lesion slightly hyperintense on T1WI and highly hyperintense on T2 WI, compared to the hypointense surrounding hepatic parenchyma. In the dynamic phases, HCC sometimes shows contrast enhancement in the arterial phase without the typical dynamic behavior on portal and delayed phases [39,40] (Figure 1). The diagnosis could be even more challenging in cases of pre-neoplastic lesions containing iron, the so-called siderotic nodules, such as regenerative or dysplastic nodules. These lesions appear hypointense in every sequence without showing hyperenhancement in arterial phase. Moreover, it has been observed that in the liver of patients affected by HH is common to find subcapsular nodules of hepatocytes free of iron or containing much less iron than the surrounding parenchyma. These nodules have been referred to as iron-free foci (IFF) and two studies conducted on large populations of patients with HH have investigated their prevalence and nature [41,42]. It has been demonstrated that IFF can represent proliferative lesions highly suggestive of pre-neoplastic or neoplastic lesions (HCC or, less frequently, malignancies other than HCC) as long as benign lesions (e.g., adenoma) and that their radiological features are not specific. Therefore, considering the high risk of HCC in HH, when MRI shows IFF, their nature should be assessed through biopsy to rule out the presence of malignancies [41]. The risk of IFF being a pre-neoplastic lesion is so high that in one patient liver transplantation was performed based solely on the presence of an IFF on MRI. The clinical suspicion of HCC was confirmed by histopathology only after transplantation [40]. Interestingly, a very few cases of HCC developed in patients with HH in absence of both cirrhosis and iron overload. In these cases, the etiology is probably multifactorial and relies on the exposure to other concomitant HCC risk factors [43].

### 4.2. Extramedullary Hematopoiesis

Extramedullary hematopoiesis (EMH) is the production of blood cells outside the bone marrow and typically occurs in cases of marrow hyperactivity, infiltration or depletion. Hyperactive bone marrow usually develops in conditions such as congenital hemolytic anemias, while marrow depletion and infiltration are more frequently observed in cases of chronic anemia with inefficient erythropoiesis, myelodysplastic syndromes, lymphoma and leukaemia [46]. EMH typically involves the liver, spleen and lymph nodes, most commonly with a pattern of microscopic infiltration. However, in rare cases, it may manifest as an FLL, which needs to be differentiated from other mass-like lesions [47]. Due to pathogenetic mechanisms and treatments (repeated transfusions) all the conditions associated with compensatory EMH previously described often develop liver iron overload [17]. To date, very few cases of focal intrahepatic EMH have been reported in the literature, and only seven describe the MRI features of such lesions, four associated to liver iron overload and three without (Table 2). In the cases reported by Wong et al. [48] and Jelali et al. [46] EMH nodules appear hyperintense compared to the surrounding liver parenchyma and isointense compared to spinal muscle, both on T1 and T2 WI. However, these lesions show different dynamic behavior after administration of Gd-chelates, resulting in heterogeneous enhancement (with maximal at 2 min) in the former case, and no enhancement in the arterial phase, with moderate enhancement in delayed phases in the latter case. The relative hyperintensity to the liver (and not to the spinal muscle) that these EMH lesions show on both T1 and T2 WI is due to the hepatic iron overload that the two patients developed as a consequence of repeated transfusions. Wong et al. [48] were the first to report the occurrence of stellate scars within intra-hepatic EMH showing hypointensity on T1 WI, hyperintensity on T2 WI and enhancement in delayed phases (5 min). Always in the context of liver iron overload, Kumar et al. [49] described a nodule of intra-hepatic EMH isointense to the surrounding liver both on T1 and T2 WI, while Belay et al. [50] reported a case of mass-forming EMH hypointense on T1 WI and isointense on T2 WI, showing heterogeneous mild arterial enhancement with washout to isointensity on portal venous and delayed phase images. Belay et al. [50] also suggested the diagnostic potential role of T2* GRE sequences in differentiating EMH from primary and metastatic hepatic lesions in diffuse hepatic iron overload. EMH may appear isointense to background liver on T2* GRE sequence, while iron-devoid adenoma, hepatoma, and hepatic metastasis appear hyperintense. According to Elsayes et al. [51] and Haidar et al. [52], the difference in signal intensity on T1 and T2 WI between the EMH nodules of the four cases described might be due to the presence of active or inactive hematopoiesis. Warshauer [53], Tamm [54] and Lee [47] described three cases of mass-like EMH lesions in normal livers that appear hyperintense on T2 WI and iso- or hypointense on T1 WI. These nodules show different behaviors in dynamic phases after administration of Gd-chelates: respectively, heterogeneous enhancement, no enhancement and homogeneous intense enhancement in the arterial phase persistent to delayed phases (5 min).

### 4.3. Adenoma (Steroid Treatment in Case of Anemia, such as Fanconi)

In subjects without underlying liver disease, the correct diagnosis of hepatic adenomas is a challenge, especially in differentiating the different subtypes (hepatocyte nuclear factor 1A-mutated, inflammatory, β-catenin-mutated and unclassified HCA) because they all typically appear hyperintense relative to the surrounding parenchyma on T2 WI, and they can range from hyper- to iso- to hypointense on T1 WI, with a common heterogeneous appearance in case of internal bleeding [55]. A signal drop out on opposed-phase T1 WI indicates the presence of fat inside the lesions [56,57]. After administration of Gd-chelates, adenomas show early enhancement in the arterial phase and become nearly isointense relative to liver on delayed phases [58]. In some types of anemia, such as Fanconi anemia, the use of steroids to stimulate erythropoietin production and iron overload due to repeated transfusions can induce the development of liver tumors (liver cell adenoma and HCC) [59]. It is important to notice that in case of diffuse hepatic iron overload, focal hepatic lesions’ signal intensity relative to the parenchyma may be impaired because of the liver hypointensity induced by susceptibility artifacts [17]. In particular liver adenomas might appear more hyperintense relative to the surrounding parenchyma than in normal liver, especially on T2 WI and in-phase T1 WI, the sequences more affected by iron overload.

### 4.4. Metastasis (Increased Risk of Colorectal Cancer)

Although there is no complete agreement in the literature, some researchers have pointed out connections between increased body iron reserves, mutations in hemochromatosis genes, and cancers other than hepatocellular carcinoma, in particular colorectal cancer (CRC) [60,61,62]. Since CRC typically spreads to the liver [63], it is important to understand how CRC liver metastasis (CRLM) appears in cases of iron overload. Due to the hepatic parenchyma hypointensity on T2 WI induced by increased iron reserves, CRLM can appear hyperintense, mimicking benign lesions, such as cysts and hemangiomas [57] with whom they also share diffusion restriction on DWI and hypointensity on the hepatobiliary phase [64]. Post-contrast sequences can be useful to differentiate those lesions, unveiling some hallmarks of secondary lesions’ dynamic behavior such as rim enhancement on late arterial phase that persists in portal venous phase on T1 WI and ill-defined enhancement of the surrounding liver parenchyma [65].

### 4.5. Liver Iron Overload Sparing

The accumulation of iron in hepatic parenchyma can be diffuse or patchy. In the latter case, it can have a nodular appearance, simulating a FLL (Figure 2). The radiologists should be aware of this possibility and be confident with the normal signal intensity of liver parenchyma in every sequence. The correlation with clinical and laboratoristic features is also mandatory for the differential diagnosis.

## 5. Conclusions

Liver iron overload is a quite common condition that can lead to the development of benign or malignant lesions. Imaging evaluation is important not only to assess and quantify hepatic iron accumulation, but also to detect focal liver lesions. Signal loss of iron overloaded hepatic parenchyma could potentially lead to misdiagnosis hence to improper patient management. A comprehensive description of MRI features of the most common FLL, summarized in Table 3, can help radiologists to reach the correct diagnosis. As the published literature is out of date, it would be useful to conduct multicentric studies and create a registry of patients with liver iron overload and liver lesions, and also to use a standardized MRI protocol. Future studies could be focused on fusion imaging, for example by integrating MRI with positron emission tomography to assess metabolic activity of FLL. Moreover, the characterization of lesions can take advantage of the application of artificial intelligence, a tool currently lacking in this field.

## Figures and Tables

**Figure 1 diagnostics-12-00891-f001:**
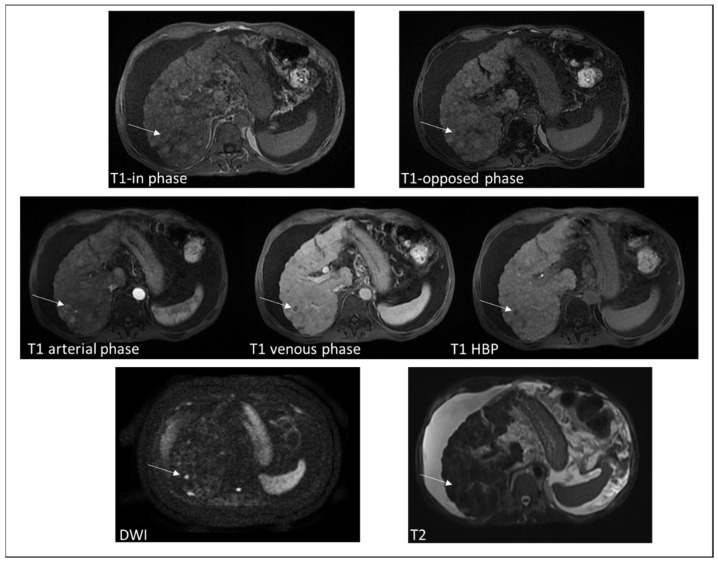
MRI images of a 55-year-old man affected by secondary hemocromatosis and HBV cirrhosis, with a nodular lesion in the VIs. The T1-in phase image (TE = 4.8 ms) did not show liver parenchyma hypointensity in comparison to T1. opposed phase image (TE = 2.4 ms) as the liver iron concentration was mild. However, in the T2 weighted image we can qualitatively appreciate a diffuse hypointensity of liver parenchyma. The lesion appeared faintly hyperintense on T2-w image, showed hypervascularization in arterial phase and wash-out in venous phase. It was hypointense in the hepatobiliary (HBP) phase and had signal diffusion restriction on diffuse weighted images (DWI). The lesion turned out to be HCC.

**Figure 2 diagnostics-12-00891-f002:**
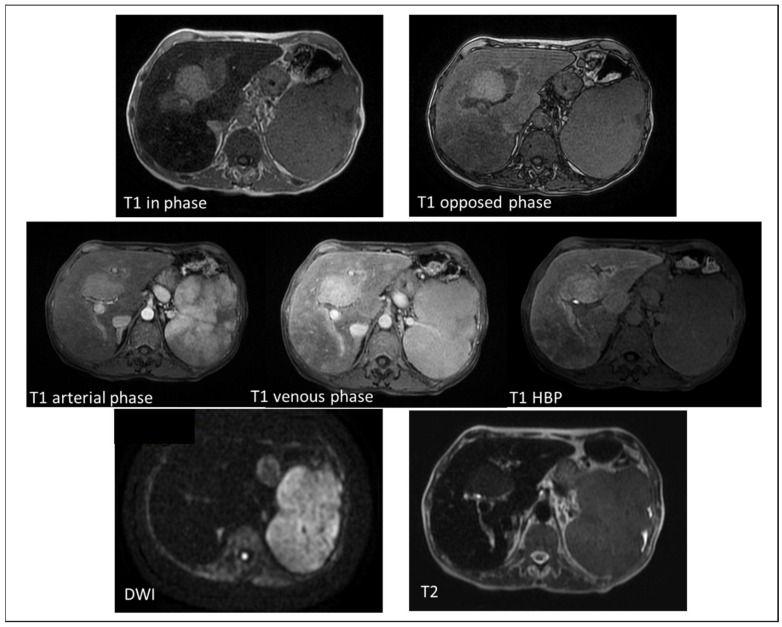
MRI imaging of a 72-year-old female with secondary hemochromatosis, showing a nodular area in the IV segment. The area, adjacent to portal bifurcation, appeared as hyperintense in comparison to the surrounding liver parenchyma both in T1 in phase and opposed phase imaging. The liver parenchyma appears less hypointense in the T1 opposed phase (TE = 2.4 ms) than in the T1 in-phase image (TE = 4.8 ms) due to the iron overload. The lesion was also slightly hyperintense in T2 weighted images without restriction on DWI sequences. After the administration of contrast agent (Gd-EOB-DTPA), it seemed to have hyperenhancement on arterial phase without wash-out in venous phase. In the hepatobiliary (HBP), it appeared hyperintense. This lesion turned out to be a nodular area of iron sparing.

**Table 1 diagnostics-12-00891-t001:** HCC in secondary hemochromatosis: review of the literature.

	Iron Overload	Cirrhosis	T1 WI	T2 WI	DWI	Underlying Condition
Barry 1968 [44]	Yes	Yes	N/A	N/A	N/A	Hereditary spherocytosis
De Tomas 1997 [38]	Yes	No (mild fibrosis)	N/A	Hyperintense to liver	N/A	None Coalminer
Chung 2003 [36]	Yes	No	Hyperintense to liver	Hyperintense to liver	N/A	MDS (RA)
Ikoma 2013 [37]	Yes	No	Hyperintense to liver	Hyperintense to liver	Hyperintense to liver	MDS (RCMD-RS)
Yamauchi 2019 [45]	Yes	No	N/A	N/A	N/A	MDS (RA)

DWI: diffusion-weighted images; MDS: myelodysplastic syndrome; RA: refractory anemia; RCMD-RS: refractory cytopenia with multilineage dysplasia and ringed sideroblasts; TACE: transarterial chemoembolization.

**Table 2 diagnostics-12-00891-t002:** Focal intra-hepatic extra-medullary hematopoiesis: review of the literature.

	Iron Overload	T1 WI	T2 WI	Gd-chelate	Additional Reports	Histopathological Findings	Other Location	Underlying Condition
Kumar 1995 [49]	Yes	Isointense to L, hypointense to M	Isointense to L, hypointense to M	N/A	On proton density WI isointense to L, central necrotic areas	N/A	Para-vertebral	β-Thalassemia
Wong 1999 [48]	Yes	Hyperintense to L, isointense to M	Hyperintense to L, isointense to M	Heterogeneous enhancement (maximal at 2 min)	Stellate scars within intra-hepatic EMH lesion (T1 hypointense, T2 hyperintense, delayed enhancement (5 min))	Hepatocytes, megakaryocytes, erythroid cell precursors, fibrous cells	No	β-Thalassemia
Jelali 2006 [46]	Yes	Hyperintense to L, isointense to M	Slightly hyperintense to L, isointense to M	Absent in arterial phase and moderate in later phases	N/A	Periportal fibrosis and sinusoid dilatation with megakaryocytes	Para-aortic and para-spinal	Sickle cell disease
Belay 2018 [50]	Yes	Hypointense to L	Isointense to L	Heterogeneous mild arterial phase enhancement with washout to isointensity on portal venous and delayed phase	Isointense to L on DWI and T2* GRE	EMH with hepatocytes and Kupffer cells iron overload	Within right main pulmonary artery	Myelodysplastic syndrome
Warshauer 1991 [53]	No	Isointense to L and M	Hyperintense to L and M	Heterogeneous enhancement	N/A	N/A	No	Unknown
Tamm 1995 [54]	No	Hypointense to L	Hyperintense to L	No enhancement on dynamic injection, delayed enhancement	On spin-density images isointense to L	EMH, enlarged Kupffer cells with “tissue paper” cytoplasm (Gaucher disease)	No	Gaucher disease
Lee 2008 [47]	No	Hypointense to L and M	Hyperintense to L and M	Homogeneous intense arterial phase enhancement persistent to delayed phases (5 min)	Hyperintense on SPIO-enhanced T2*WI	Megakaryocytes, erythroid and myeloid precursors, histiocytic cells (CD68 IHC weaker than normal). No Kupffer cells	No	Essential thrombocythemia transformed into idiopathic myelofibrosis

L = liver; M = spinal muscle; ICH = Immunohistochemistry; SPIO = Superparamagnetic Iron Oxide. N/A = not assessed.

**Table 3 diagnostics-12-00891-t003:** Focal liver lesions’ appearance in diversely weighted sequences compared to the intensity of the surrounding iron overloaded liver parenchyma.

	T1 WI	T2 WI	T2*	DWI	Arterial Phase	Portal-Delayed Phases
HCC	+	+	+	+	+/0	0
EMH	+/0/−	+/0	0	N/A	+/0	+/0
Adenoma	+	+/0	+	N/A	+	+/0
Liver mestastasis		+	+	+	Rim enhancement	Ill-defined enhancement of surrounding parenchyma
Liver iron overload sparing	+	+	+	0	+	+

“+” stands for hyperintense; “−“ stands for hypointense; “0” stands for isointense. HCC = Hepatocellular Carcinoma; EMH = Extramedullary Hematopoiesis; WI = weighted image; DWI = Diffusion-Weighted Image.

## Data Availability

Not applicable.

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
