# Peer review of "MRI Appearance of Focal Lesions in Liver Iron Overload"

_diagnostics, 2022, doi:10.3390/diagnostics12040891_

Round 1
Reviewer 1 Report
Dear Authors:
I have read with pleasure your interesting review. The presence of iron overload in the liver may make it difficult to do a correct diferential diagnosis between different focal liver lesions. A review about MRI appearance of focal lesions in liver with iron overload is of great interest and may help us in clinical practice.
Comments and corrections:
-Title: I think that "Background" should be deleted. "MRI appearance of focal lesions in liver iron overload" is better. line 3.
-abstract: Ok. To date,... is used twice, and perhaps "Nowadays" would be a better term with: "Nowadays, Magnetic Resonance Imaging (MRI)..."-line 12.
-3.Imaging diagnosis...
3.2.2 signal intensity ratio
line 183-4: ...and tends to overestimate mild and moderate iron deposition (5).
This is true with the Rennes University method, but not with theSEDIA`s model proposed by Alustiza et al (19) that uses the same metod but only with two sequences and a different mathematical formula to calculate LIC.
The results of SEDIA's model are better correlated with R2* results and with LIC measured on liver biopsies.
*Correct it.
The most significant advantages of SIR methos are their accesibility, being feasible on every machine in the world (21).
*Comment it.
Interesting review.
Author Response
Dear Marcin Biedron
Assistant Editor
Diagnostics
Manuscript ID: diagnostics-1638896
Title: MRI appearance of focal lesions in liver iron overload
On behalf of all the authors, I would like to thank you for your consideration of this paper.
In the revised manuscript you will find the underlined changes made in response to the Reviewers.
Please find below a point-by-point response to the Reviewers’ comments.
Finally, I would like to thank the Editor and the Reviewers for their comments that helped us to increase the value of our paper.
Best regards
Anna Pecorelli, MD PhD
Reviewer #1
I have read with pleasure your interesting review. The presence of iron overload in the liver may make it difficult to do a correct differential diagnosis between different focal liver lesions. A review about MRI appearance of focal lesions in liver with iron overload is of great interest and may help us in clinical practice.
Answer (A): Thank you for your comment. We appreciate that you find this review helpful.
Comments and corrections:
Question (Q): Title: I think that "Background" should be deleted. "MRI appearance of focal lesions in liver iron overload" is better. line 3.
A.: Thank you for this suggestion. We deleted “background” from the title.
Q.: abstract: Ok. “To date,...” is used twice, and perhaps "Nowadays" would be a better term with: "Nowadays, Magnetic Resonance Imaging (MRI)..."-line 12.
A.: Thank you, we have corrected the repetition.
Q: 3. Imaging diagnosis...
3.2.2 signal intensity ratio
line 183-4: ...and tends to overestimate mild and moderate iron deposition (5).
This is true with the Rennes University method, but not with the SEDIA’s model proposed by Alustiza et al (19) that uses the same method but only with two sequences and a different mathematical formula to calculate LIC.
The results of SEDIA's model are better correlated with R2* results and with LIC measured on liver biopsies.
*Correct it.
A.: Thank you for this important suggestion. We have updated the pertinent section with SEDIA’s model, its advantages and the corresponding citation.
Q.: The most significant advantages of SIR methods are their accessibility, being feasible on every machine in the world (21).
*Comment it.
A.: We agree with this sentence that has now been added in the relative paragraph.
Interesting review.
A.: Thank you again for your kind comment.

Reviewer 2 Report
The paper deals with focal liver lesions in patients with different forms of iron overload. It comprehensively presents the background of iron metabolism and excessive iron storage due to various reasons. However, the intention of this manuscript remains unclear. It does not contain original research, but for a review its concept lacks a systematic approach.
Comments in detail:
P 3, l 105: We consider the statement ‘signal intensity decay proportional to the iron overload’ rather confusing. The relaxation times mentioned at the beginning of the sentence are not proportional to iron overload, but the transversal relaxation rates, the inverse of transversal relaxation times, as authors state in section 3.2.1. Please, be more precise here, too.
P 3, l 107: We recommend to use the term ‘opposed phase’, not ‘out-of-phase’, since it indicates that fat and water signals point in opposite direction causing subtraction of signals. ‘Out-of-phase’ only indicates that signals are not aligned.
In- (IP) and opposed-phase (OP) images give information mainly about tissue fat content. Since the in-phase condition occurs at several TEs at a certain TE increment (about 4.7 ms at 1.5 T), you cannot generally state that IP images have a larger TE than OP images – e.g., in-phase occurs at 4.7 ms, opposed-phase not only at 2.35 ms but also at 7.1 ms which is larger than 4.7 ms. The crucial point for quantification of iron concentration is to acquire images at different TEs to evaluate the signal decay due to T2* (or T2) relaxation. Gradient-echo sequences, which have - in contrast to spin-echo - no refocusing pulse, show signal oscillations (alternating OP- and IP-conditions with increasing TE) due to different resonance frequencies of fat and water in tissues containing both components, like the liver.
P 3, l 136: Meanwhile, a variant of St. Pierres method is available with a reduced TR of 1000 ms shortening the acquisition time.
P 4, l 148, also l 151: The name of this gentleman is Wood, but not Wool.
P 4, l 149: In Woods publication, a TR of 25 ms is mentioned.
P 4, l 172: The first work on SIR at 3T was published by Wunderlich et al. (https://pubmed.ncbi.nlm.nih.gov/27299667/)
P 4, l 175: The SIR-LIC-curve can be linearized only approximately and only in sections, depending on the protocol used. It can easily be shown that there is a linear dependence of the logarithm of SIR on LIC. (https://pubmed.ncbi.nlm.nih.gov/27299667/)
P 4, l 182: Rose et al. published an extension to the SIR method suitable for high LIC. (https://pubmed.ncbi.nlm.nih.gov/16608501/)
P 5, l 207: Since T1w images are acquired at a rather short TE, we assume hypointense liver parenchyma only at high LIC levels in T1w. Please, comment.
P 7, Table 2: What was the strategy and/or criteria to select these papers? Please, explain.
P 8, Fig. 1: We consider it confusing that in the opposed-phase image, which shows the difference of fat and water signal, the liver appears brighter than in the in-phase image. Giving the TEs and maybe a short comment in the figure caption would be helpful. What is shown in the rightmost image of the second row? Please, annotate also this subimage.
P 9: At the end of the fourth section (Focal liver lesions …) we recommend a table summarizing the different lesions mentioned and their appearance in the different MR protocols.
Author Response
Dear Marcin Biedron
Assistant Editor
Diagnostics
Manuscript ID: diagnostics-1638896
Title: MRI appearance of focal lesions in liver iron overload
On behalf of all the authors, I would like to thank you for your consideration of this paper.
In the revised manuscript you will find the underlined changes made in response to the Reviewers.
Please find below a point-by-point response to the Reviewers’ comments.
Finally, I would like to thank the Editor and the Reviewers for their comments that helped us to increase the value of our paper.
Best regards
Anna Pecorelli, MD PhD
Reviewer #2
The paper deals with focal liver lesions in patients with different forms of iron overload. It comprehensively presents the background of iron metabolism and excessive iron storage due to various reasons. However, the intention of this manuscript remains unclear. It does not contain original research, but for a review its concept lacks a systematic approach.
A.: We apologize for the unclarity of the manuscript. We hope that after our modifications and answers to the precious reviewers’ comments it will be clearer and more accurate.
Comments in detail:
Q.: P 3, l 105: We consider the statement ‘signal intensity decay proportional to the iron overload’ rather confusing. The relaxation times mentioned at the beginning of the sentence are not proportional to iron overload, but the transversal relaxation rates, the inverse of transversal relaxation times, as authors state in section 3.2.1. Please, be more precise here, too.
A.: Thank you for your comment, we modified our sentence in order to avoid any confusing statement. In agreement with [1], the new sentence is: “The liver iron deposits introduce inhomogeneities in the main magnetic field due to the superparamagnetic effect of the iron ions. This leads to the reduction of the relaxation time of hydrogen nuclei and signal intensity with increasing amounts of iron.”.
- Mazé, J.; Vesselle, G.; Herpe, G.; Boucebci, S.; Silvain, C.; Ingrand, P.; Tasu, J.-P. Evaluation of Hepatic Iron Concentration Heterogeneities Using the MRI R2* Mapping Method. Eur. J. Radiol. 2019, 116, 47–54, doi:10.1016/j.ejrad.2018.02.011.
Q.: P 3, l 107: We recommend to use the term ‘opposed phase’, not ‘out-of-phase’, since it indicates that fat and water signals point in opposite direction causing subtraction of signals. ‘Out-of-phase’ only indicates that signals are not aligned. In- (IP) and opposed-phase (OP) images give information mainly about tissue fat content. Since the in-phase condition occurs at several TEs at a certain TE increment (about 4.7 ms at 1.5 T), you cannot generally state that IP images have a larger TE than OP images – e.g., in-phase occurs at 4.7 ms, opposed-phase not only at 2.35 ms but also at 7.1 ms which is larger than 4.7 ms. The crucial point for quantification of iron concentration is to acquire images at different TEs to evaluate the signal decay due to T2* (or T2) relaxation. Gradient-echo sequences, which have - in contrast to spin-echo - no refocusing pulse, show signal oscillations (alternating OP- and IP-conditions with increasing TE) due to different resonance frequencies of fat and water in tissues containing both components, like the liver.
A.: Thank you for your comment, we agree with you and we changed the expression “out of phase” with “opposed phase” all over the manuscript.
Q.: P 3, l 136: Meanwhile, a variant of St. Pierre’s method is available with a reduced TR of 1000 ms shortening the acquisition time.
A.: Thank you for your suggestion. We agree with you about the importance of St. Pierre’s method’s variant because it helps to reduce the acquisition time. We added it and the corresponding citation in the relative sections.
Q.: P 4, l 148, also l 151: The name of this gentleman is Wood, but not Wool.
A.: We apologize for this misspelling. The name has now been corrected.
Q.: P 4, l 149: In Woods publication, a TR of 25 ms is mentioned.
A.: Thank you for this comment. We corrected the TR as mentioned in Wood’s publication
Q.: P 4, l 172: The first work on SIR at 3T was published by Wunderlich et al. (https://pubmed.ncbi.nlm.nih.gov/27299667/)
A.: Thank you for this clarification. We updated the citations in the pertinent section.
Q.: P 4, l 175: The SIR-LIC-curve can be linearized only approximately and only in sections, depending on the protocol used. It can easily be shown that there is a linear dependence of the logarithm of SIR on LIC. (https://pubmed.ncbi.nlm.nih.gov/27299667/)
A.: Thank you for this clarification. We added a comment in the pertinent paragraph and the citation in the bibliography.
Q.: P 4, l 182: Rose et al. published an extension to the SIR method suitable for high LIC. (https://pubmed.ncbi.nlm.nih.gov/16608501/)
A.: Thank you for this suggestion. We added the extension of the SIR method for high LIC proposed by Rose et al, updating the bibliography.
Q.: P 5, l 207: Since T1w images are acquired at a rather short TE, we assume hypointense liver parenchyma only at high LIC levels in T1w. Please, comment.
A.: Thank you for this comment that we fully share. Liver parenchyma hypointensity on short TE T1-weighted images becomes evident when there is a high hepatic iron accumulation and the qualitative assessment of liver iron concentration might be misleading in case of mild or moderate iron overload. This is the reason why iron quantification methods were introduced, not only for the diagnosis but also to assess the response to therapy.
Q.: P 7, Table 2: What was the strategy and/or criteria to select these papers? Please, explain.
A.: To select these papers we conducted a systematic research on Pubmed using the following keywords “intra hepatic extramedullary hematopoiesis AND MRI” and we compared the cases with and without liver iron overload in order to better understand the MRI appearance of this type of focal liver lesions.
Q.: P 8, Fig. 1: We consider it confusing that in the opposed-phase image, which shows the difference of fat and water signal, the liver appears brighter than in the in-phase image. Giving the TEs and maybe a short comment in the figure caption would be helpful. What is shown in the rightmost image of the second row? Please, annotate also this subimage.
A.: Thank you for your comment. We modified the figure caption and legend according to your suggestion.
Q.: P 9: At the end of the fourth section (Focal liver lesions …) we recommend a table summarizing the different lesions mentioned and their appearance in the different MR protocols.
A.: Thank you for your useful suggestion. We added a summarizing table of the different lesions mentioned and their appearance in different MRI sequences.

Reviewer 3 Report
The manuscript presents a critical review on the evaluation of MRI findings of some of the most common hepatic focal lesions in patients with liver iron overload.
It was found that a comprehensive description of MRI features of the most common focal liver lesions can help radiologists reach the correct diagnosis.
I find the topic interesting and being worth of investigation and the document is well strucutred, organized, fluidly written, has enough background information, the methodology followed is clearly explained, the results are clearly presented and support the conclusions.
Although I propose the following suggestions:
- Abstract requires structuring such as: problem, motivation, aim, methodology, main results, further impact of those results.
- keywords should be in alphabetical order.
- Further research suggestions should be made at conclusion section
This review would benefit of being a systematic review using an approach such as PRISMA, would make it repeatable.
Author Response
Dear Marcin Biedron
Assistant Editor
Diagnostics
Manuscript ID: diagnostics-1638896
Title: MRI appearance of focal lesions in liver iron overload
On behalf of all the authors, I would like to thank you for your consideration of this paper.
In the revised manuscript you will find the underlined changes made in response to the Reviewers.
Please find below a point-by-point response to the Reviewers’ comments.
Finally, I would like to thank the Editor and the Reviewers for their comments that helped us to increase the value of our paper.
Best regards
Anna Pecorelli, MD PhD
Reviewer #3
The manuscript presents a critical review on the evaluation of MRI findings of some of the most
Common hepatic focal lesions in patients with liver iron overload. It was found that a comprehensive description of MRI features of the most common focal liver lesions can help radiologists reach the correct diagnosis. I find the topic interesting and being worth of investigation and the document is well structured, organized, fluidly written, has enough background information, the methodology followed is clearly explained, the results are clearly presented and support the conclusions.
A.: Thank you for your kind comment.
Although I propose the following suggestions:
Q.: Abstract requires structuring such as: problem, motivation, aim, methodology, main results, further impact of those results.
A.: Thank you for this comment. As long as the paper is a review and not an original article, it would be difficult to structure the abstract as you suggest. Hence, we organized the abstract in line with the instructions for authors provided by the journal.
Q: keywords should be in alphabetical order
A.: Thank you for your suggestion. The keywords order has now been corrected in an alphabetical way.
Q.: Further research suggestions should be made at conclusion section.
A.: Thank you for this important suggestion that improves our work. We added a comment about future research at the end of the conclusions section.
Q.: This review would benefit of being a systematic review using an approach such as PRISMA, would make it repeatable.
A.: Thank you for your comment. We agree with you that this review would benefit from the application of PRISMA approach. However, we decided not to apply it at this time as there are still scant published data. We think the use of PRISMA approach will be feasible when more cases and structured studies with larger samples will be available in the literature, thus making a systematic review more proper.

Round 2
Reviewer 2 Report
We acknowledge the improvements to the manuscript. However, there are still some points which have to be addressed. We regret that we noticed some of them only on second reading.
P 3, l 111: It is not quite precise to talk about in-phase and opposed-phase ‘sequences’. All are Gradient Echo sequences and differ only in the acquisition parameter ‘echo time’ (as lined out e.g. in Hernando et al., R2* estimation using "in-phase" echoes in the presence of fat: the effects of complex spectrum of fat. JMRI 2013). It would be appropriate to name it ‘protocols’ instead of ‘sequences’.
Same line: The statement that in general the in-phase condition occurs at larger echo times than opposed phase has not been corrected despite our previous comments. Please, explain or correct.
P 3, l 149: Maybe authors want to express that St. Pierres method is not able to address hepatic fat content since fat and water signal is refocused when using spin echo sequences. Therefore, fat cannot be quantified with spin-echo. 'Fat saturation', which is feasible also with spin-echo, would be of limited use only.
P 5, l 252: Since HCC is hypodense in T1w compared to healthy liver parenchyma, we expect only weak contrast to highly iron overloaded liver in T1w. In T2w, on the other hand, contrast to iron overloaded tissue should be elevated due to two reasons: lack of iron and increased T2(*) because of malignancy. Please, comment.
P 6, l 314: Were the acquisition parameters for images in this figure, namely TEs for in- and opposed-phase images, the same as given in Fig. 2?
Author Response
Dear Marcin Biedron
Assistant Editor
Diagnostics
Manuscript ID: diagnostics-1638896
Title: MRI appearance of focal lesions in liver iron overload
On behalf of all the authors, I would like to thank you again for your consideration of this paper.
In the second version of the revised manuscript you will find the underlined changes made in response to the Reviewer.
Please find below a point-by-point response to the Reviewer's comments.
Finally, I’d to thank the Editor and the Reviewer for their comments that helped us to increase the value of our paper.
Reviewer comment
We acknowledge the improvements to the manuscript. However, there are still some points which have to be addressed. We regret that we noticed some of them only on second reading.
Q: P 3, l 111: It is not quite precise to talk about in-phase and opposed-phase ‘sequences’. All are Gradient Echo sequences and differ only in the acquisition parameter ‘echo time’ (as lined out e.g. in Hernando et al., R2* estimation using "in-phase" echoes in the presence of fat: the effects of complex spectrum of fat. JMRI 2013). It would be appropriate to name it ‘protocols’ instead of ‘sequences’.
A: Thank you for your suggestion that improves the quality of our manuscript. We replaced the term “sequences” with “protocols”.
Q: Same line: The statement that in general the in-phase condition occurs at larger echo times than opposed phase has not been corrected despite our previous comments. Please, explain or correct.
A: Thank you for your comment, we apologize for our oversight. Now we have corrected the statement.
Q: P 3, l 149: Maybe authors want to express that St. Pierres method is not able to address hepatic fat content since fat and water signal is refocused when using spin echo sequences. Therefore, fat cannot be quantified with spin-echo. 'Fat saturation', which is feasible also with spin-echo, would be of limited use only.
A: thank you for this clarification. We modify the sentence accordingly.
Q: P 5, l 252: Since HCC is hypodense in T1w compared to healthy liver parenchyma, we expect only weak contrast to highly iron overloaded liver in T1w. In T2w, on the other hand, contrast to iron overloaded tissue should be elevated due to two reasons: lack of iron and increased T2(*) because of malignancy. Please, comment.
A: thank you for this comment. We agree with you: the HCC signal intensity is weak compared to the signal intensity of highly liver overloaded parenchyma on T1 WI. On the contrary on the T2 WI the contrast between HCC is high for the reason you explained. Ww corrected the sentence.
Q: P 6, l 314: Were the acquisition parameters for images in this figure, namely TEs for in- and opposed-phase images, the same as given in Fig. 2?
A: Thank you for your question. Yes, the acquisition parameters are the same for the images of both figures. We added them in the caption of Figure 1.
